# Ernesto Cardenal: A Latin American Liberation Mystic

**Marcela Raggio** [1,2]

1 CONICET, Godoy Cruz, Buenos Aires 2290, Argentina; marcelar@ffyl.uncu.edu.ar
2 Facultad de Filosofía y Letras, Universidad Nacional de Cuyo, Mendoza M5500, Argentina

**Abstract:** This paper explores mysticism as seen in Ernesto Cardenal's *El Evangelio en Solentiname* (*The Gospel in Solentiname*), aiming at both defining Cardenal as a revolutionary and a traditional mystic, shaped by Thomas Merton's influence and by Latin American political circumstances. Mysticism is usually defined as individual contemplation of God, immediate and unmediated. Yet, in the context of Latin American 20th-century struggles for liberation, mysticism became contemplation of God while the individual is committed to the community. This perspective is studied in Cardenal's book, supported with his memoir *Las ínsulas extrañanas* (*The Strange Islands*), to show that Cardenal is a mystic, notwithstanding his political commitment, or precisely because of that. The theoretical background draws notions from liberation theology and liberation philosophy. Paradoxically, in spite of its revolutionary claims, Cardenal's *The Gospel in Solentiname* can be seen in the line of traditional mysticism, in its challenge of power from the margins and its presentation of alternative modes of communicating with the divine.

**Keywords:** Ernesto Cardenal; mysticism; liberation





## 1. Introduction

Latin American philosophy of liberation understands that the universal should be related to the particular and that philosophical thinking should be located. In Latin America, the peripheral *locus* of philosophy transforms it necessarily into liberation philosophy. The situated contributions of such a philosophy, anchored in a concrete reality, at the same time transcend it, because oppression can be found anywhere and liberation is necessary everywhere.

From the intersection of the theology and philosophy of liberation as defined by Enrique Dussel, this paper reads *El Evangelio en Solentiname* (*The Gospel in Solentiname*) ((1975) 1979) and Cardenal's praxis narrated in *Las ínsulas extrañas* (*The Strange Islands*)[1] (Cardenal 2002) as a manifestation of the Nicaraguan poet's mysticism. This paper aims at showing that Cardenal's commitment to the liberation of his country is a sign of his *mysticism*, in the context of both the theology and philosophy of liberation, in a continent and a historical period marked by social injustice. Just as liberation philosophy starts in the concrete Latin American (marginal) reality, as a situated system of thought, so its precedent liberation theology marks the pre-eminence of evangelical praxis (which by definition is situated) over dogma (which is universal). In this frame of ideas, the works and praxis of Nicaraguan poet, priest and activist Ernesto Cardenal (1925–2020) can be studied as the epitome of liberation ideas.

Ernesto Cardenal was born in a well-to-do family in Nicaragua. He had the chance to study at Columbia University in the United States, and after his religious conversion, he entered the Trappist monastery in Gethsemani, Kentucky, where he was a novice under Thomas Merton (1915–1968) in the late 1950s. Due to a health problem, Cardenal left the abbey and attended seminary in Mexico and then in Colombia, to be ordained a priest in his native Nicaragua in 1965. The country was at the time under the dictatorship of Anastasio Somoza Debayle (son of Anastasio Somoza García and brother of Luis Somoza Debayle, both of whom preceded Anastasio as dictators). Therefore, from 1937 until 1979, Nicaragua

was dominated by the Somozas, who controlled the country with tyranny and oppression. Repression, persecution, kidnapping and torture were the methods used by the Somozas to control a country stricken by poverty, unequal distribution of wealth and suppressed freedoms. Political repression, social tension and socio-economic crisis, together with terrorism against peasants and religious and political organizations and the suspension of constitutional guarantees in 1974, fueled the anti-somocista feeling and the support that the Sandinista National Liberation Front (FSLN for its name in Spanish) would obtain in the late 1960s and 1970s, leading to the triumph of the Sandinista Revolution in 1979 (González Arana 2009).

Prompted by his mentor Thomas Merton and his own understanding and practice of the theology of liberation, Cardenal started his community at Solentiname in 1965, in which we may see the mystic experience in the Latin American context as understood by the theology of liberation.

Londoño (2021) points out that liberation theology highlights that mysticism is contextual (p. 144) and that God's mystery is manifested in actions (p. 145). Thus, faith is a contemplative act accompanied by social revindication in a world marked by structural oppression (p. 147). Contemplation cannot be separated from the praxis of political action. This paper presents such notions based on the contributions of Leonardo Boff, Enrique Dussel and Juan Carlos Scannone. Special attention is paid to Pablo Mello's consideration of the moments of liberation in mystic silence.

Boff (1991) stresses that the mystical experience is, of course, spiritual, yet it is historically located: "Any spiritual experience means meeting a new and defying face of God, which stems from the great challenges of historical reality [ . . . ] God appears [ . . . ] as a *meaningful event*, as a sign of *hope, of absolute future for man and his history. This situation fosters a proper and typical experience of God's mystery.*"[2] (p. 56, italics in the original). In the Latin American context, the historical situation that reveals God's face is the need for liberation of the poor, in whom God has manifested His own "*demands of solidarity, of identification, of justice and dignity.*" (p. 56, italics in the original). Due to the intrinsic connection between the mystical experience and its historicity, "*heaven is not an enemy of earth; it begins on earth already.* [ . . . ] This is not mere theology. It is the life and mysticism of many Christians." (p. 67, italics in the original). Such a connection with social and historical circumstance is unavoidable in the theology of liberation. García (1987) defines it as "that form of reflection that attempts to discern the religious significance of sociopolitical struggles in which the poor are engaged as they free themselves of their present state of political domination and economic exploitation" (p. 7).

In order to understand Cardenal's mysticism in the context of Latin American theology of liberation, we follow Dussel (1995), who considers that Solentiname can be seen as a sample of the sixth period in the history of theology in Latin America. Dussel affirms that "the contemplative movement generated by the liberation process emerges as a new type of 'spirituality'" (p. 141). He presents Cardenal as one of the representatives of this new spirituality: "Ernesto Cardenal, trappist with Thomas Merton, creator of a new way of monastic life in Solentiname, and especially in his revolutionary commitment both in his *Psalms* and in *The Sanctity of Revolution*, since 'sanctity' is not just Christian, but of those who die for love of their brethren." (Dussel 1995, pp. 141–42).

Dussel's analysis should be central to any approach to liberation theology, both because he was one of its main theorists, and because at present he continues to revisit his contributions to contemporary developments of liberation movements in the twenty-first century. While Dussel, himself an Argentine exiled in Mexico since the mid-1970s, explains the South American origin of liberation theology, he maintains that the political circumstances of Latin American countries during the 1960s and 1970s led to a type of theology produced by people working in teams, in reflection "centers" away from ecclesial centers (pp. 158–59). Of special interest to our paper is his appreciation of the Nicaraguan situation, when due to several circumstances, the theology of liberation expanded from the South Cone to the rest of Latin America, particularly to Central America, Mexico and

the Caribbean (p. 159). In this context, according to Dussel, "the Sandinist revolution is central to this period. All Central America begins a revolutionary process of Christian commitment and resistance against repression." (p. 159). Particularly in Nicaragua, "the revolutionary process calls for a theological clarification of faith. Sandinista ideology is not a mere repetition of what was already known." (p. 160). The Nicaraguan Church is indissolubly linked to the revolutionary process, thus producing a cultural revolution as well (pp. 161–62).

In addition, he follows a parallel study of the philosophy of liberation in his 1975 volume (2013). Dussel maintains that "liberation is the praxis which subverts the phenomenological order and drills it towards a metaphysical transcendence which is a total criticism of the established, fixed, normalized, crystallized, dead order." ([Dussel 2013](), p. 80). Liberation implies an ethics, says Dussel, in which listening to the other, to his just protest, may call into question the basis of the system. Thus, in order to listen, one has to become silent, and silence (listening) is a form of respect. Then the listener, by learning about the other, acquires a responsibility for him (social responsibility, because there is a need for a new system). Dussel goes on to explain that in order to establish a new system, the old one must pass away, as history shows. Therefore, "Every moment of passage is agonic; thus liberation is equally agonic of the old in order to achieve a fertile birth of the new, of what is just." (p. 83). Liberation also implies considering the real face of the other, instead of his mask as a peasant, worker, etc., and in this lifting off of masks, liberation involves not just listening, but also *seeing* the other: "It is a challenging seeing, which promotes mercy, justice, rebellion, revolution, liberation." (p. 85). Dussel's categories of liberation are *praxis*, or the mere creation of the new order (p. 86), and *ethos*, or solidarity and love for the oppressed in their own dignity (p. 87). This whole ethos stems from the solidarity of all involved in the liberation process, who call for "real justice, that it, subversive or subverting of the unjust established order." (p. 87). For Dussel, the process of liberation may even reach subversive illegality, because it works against the system in order to establish a new one (p. 88).

Since our analysis is based on written texts, we should consider another notion that Dussel develops in his *Filosofía de la liberación*: "A semiotics of liberation must describe the process of passing from an existing sign system to a new order born from destruction of and going beyond the old order." (p. 150).

In the liberation process, the analectic moment is fundamental. For Dussel, "Offering even one's own life in order to fulfill the requirements of protest, and launching out into praxis for the oppressed, is part of the analectic moment. Theory is not enough in analectics." (p. 186). Since praxis is central to liberation, it involves meeting and interacting with the other, and that praxis of liberation calls for a new way of expressing it, a new type of discourse. As Dussel puts it, in order to be radical, the discourse of liberation "must have a different starting point, must think of other themes, must get to different conclusions and with a different method." (p. 200). Such a discourse must think of what was unthought of before, that is, liberation of the oppressed.

Our approach to Cardenal's texts, then, is based on Dussel's definition: "Philosophy of liberation is a pedagogical operation, from a praxis that is established in the proximity teacher-disciple, thinker-people." (p. 205). Because the Nicaraguan poet understands and takes the risks of a philosophy of liberation and articulates theology with his people, the people of Solentiname, in a specific context that calls for a clarification of faith in praxis, and in a specific context, as theologians of liberation stress must be, he is a Latin American mystic both in his writings and in his actions.

[Scannone]() ([2009]()) makes it clear that even if the theology of liberation and philosophy of liberation have been connected by some critics, the philosophy of liberation is autonomous in its reflection (p. 61). He points out that like any other philosophical system, this one is based on reality. What distinguishes it from other systems of thought is that the philosophy of liberation considers the socio-historical reality of the victims of injustice in Latin America (p. 68), which explains why the *praxis* of liberation is central to it. While Scannone writes

about the philosophy of liberation, some of the notions he explores can be applied to our understanding of the theology of liberation in Cardenal's works. Scannone (2009) explains that "victims not only challenge and question philosophers and their activity, but they also teach the basic human wisdom which is frequently born out of injustice lived by each one and by others. In our context it is Latin American popular wisdom." (p. 69). We may argue that just as philosophy learns from the victims, the poor, so does the theology of liberation, as we prove later in our analysis of Cardenal's works.

Mella (2009) studies mysticism in Latin America and defines the mystic experience as follows: "Mystic experience [ . . . ] has traditionally been presented as an intense experience of God, of the sacred, or of the totality of worldly experience, which challenges the social order, shaking subjectivity, including the relation with one's own body, to arrive at the ineffable." (p. 366). Mella's paper situates the mystic experience in the Latin American context, thus becoming a useful instrument to analyze Ernesto Cardenal's texts and project. Mella's approach to mysticism in Latin American explores six counter-theses. First, by stating that there is not an essential form of the mystic phenomenon, he clarifies that words are not enough to transmit the experience, which calls for novel forms of poetic expression, of dissatisfaction with existing reality. This, in turn, may be considered dangerous by the establishment, as has happened throughout the history of mysticism (pp. 372–73). Mella then goes on to explain his second counter-thesis, related to mystical silence. Far from being politically irresponsible, silence in liberation mysticism implies taking on the reality of the oppressed (p. 380). Mella's analysis is based on González Buelta's poetry, which helps him identify six moments in liberation mystic silence:

1. "Loss of meaning of known discourse
2. Invitation to search new social meanings
3. Understanding that God still walks next to the people
4. Looking back on history to understand that God has always manifested Himself in liberation processes, while never ceasing to be transcendent
5. A new way of understanding one's own commitment to freedom as the fragile manifestation of what really matters, and
6. At the end of the journey (the mystic journey), a deep peace enables the mystic to accept he/she is a regular human being, whose achievements in the struggle for liberation are not a result of his/her own efforts, but a humble participation in divine life, which is given graciously everywhere, without asking for anything in return" (p. 382).

This leads to Mella's next counter-thesis, by which the Christian liberation mystic is able to see God's face in the oppressed and commits him/herself to struggling against situations of violence (p. 389).

Based on the theoretical framework presented above, we present the following hypothesis: Ernesto Cardenal presents the traits of Latin American liberation mysticism in his journal *The Strange Islands* and his book *The Gospel in Solentiname*. The analysis of his discourse of liberation, to use Dussel's expression, shows the intersection of the political stand of liberation theology and mysticism in Latin America, as a way to narrate political engagement (Londoño 2021, p. 139).

## 2. Community and Gospel in Solentiname: The Way of Liberation

### 2.1. A New Society in Solentiname

By the time Cardenal was ordained in August 1965, his project of setting up a monastic foundation in the Nicaraguan archipelago of Solentiname had been accepted by his superiors, as he states in *The Strange Islands*:

> I was ordained in Nicaragua by Monsignor Barni, bishop of Rio Chontales and Rio San Juan, which included Solentiname as well. Because he agreed to my foundation in Solentiname, as long as Rome would accept it (and Rome's approval was that the papal nuncio approved it). (Cardenal 2003, p. 68)

Prior to the establishment of the Solentiname community, Cardenal had been studying at the seminary in Colombia, where the changes the church was undergoing were strongly felt. During his time at the seminary, Cardenal formed part of the Halleluiahs, a group of seminarists who felt the mystical essence of their vocation. Cardenal points out that the group was resented by the more clericalist priests and students and the seminary, who were "viscerally anti-mystics" and who felt they "threatened their estatu quo" (Cardenal 2003, p. 33). Throughout history, mystics have been resented in like manner, and Cardenal was not an exception. In his seminarian days, as recounted in *The Strange Islands*, the various aspects that would combine and produce Cardenal's later mysticism in connection with the Nicaraguan liberation process were already at play. His support of liberation is wholistic, in that Cardenal does not see political or economic freedom separate from man's spiritual side. As he said in a 1974 interview with Ronald Christ, "I am not interested in an economic liberation of man without the liberation of the whole man" (Christ 1974). López-Baralt (2010) presents the mystic experience as informing all of Cardenal's activities:

> I understand that all the poet's [Cardenal's] activities, his entering the Trappist abbey, his studies for the priesthood in Colombia, his politization in Solentiname, his commitment to Sandinismo, his being Minister of Culture, his participation in collective alphabetization, his poetic tutoring of children with cancer, his sculpting, are all the exterior manifesttation of an ad intra spiritual process which has marked him for ever. (López-Baralt 2010, p. 11)

In this article, we concentrate on the first items in López-Baralt's enumeration, and in them we can detect the deep connection between liberation and mysticism, inextricably joined in his praxis and in his discourse. What makes Cardenal a mystic in the Latin American tradition is his deep commitment with his reality, after a journey both in a geographical and an ideological sense.

The discourse of liberation as shown in *The Strange Islands* presents at least four strands: Thomas Merton's influence, Cardenal's interest in Central American indigenous culture, the impact of revolutionary thought and praxis, and the experience of the mystical union.

2.1.1. Thomas Merton's Influence

Cardenal had met Merton at Gethsemani in 1957, when he entered the abbey as a novice. After his conversion to Catholicism, very much like that of Merton himself, Cardenal started his purification in rural Kentucky, unknowingly getting ready for the mystic way to be completed in the Latin American context. When Cardenal left Gethsemani in 1959, his friendship with Merton would continue in epistolar form and in a visit of Cardenal to the abbey after he was ordained. Merton's manifold influence has been pointed out by Santiago Daydí-Tolson (2003), who states, "This social responsibility of catholic intellectuals, and even of contemplatives, is a central aspect of Merton's influence on Cardenal" (Daydí-Tolson 2003, p. 23). Daydí-Tolson goes on to say, "Another aspect of Merton which would have a strong influence on Cardenal's religious thought and political action is his conviction of the need to establish a dialogue with left-wing politics as an alternative to social injustice" (Daydí-Tolson 2003, p. 23). Cardenal's interest in native American peoples is also connected to Merton's guide, as Daydí-Tolson explains: "Merton approves of and promotes Cardenal's interest in writing about the native cultures of th Americas, because they have a spiritual value superior to that of Western culture." (p. 23). Finally, according to Daydí-Tolson, "Solentiname, with its strong mark of an artistic community, free from the rules of a traditional monastery, fulfills Merton's idea of what an authentic monastic community should be." (Daydí-Tolson 2003, p. 25).

All these aspects that Daydí-Tolson presents reflect the characteristics of Latin American mysticism, though, paradoxically, or coincidentally, they are derived from Merton's influence as well. Cardenal himself recognizes that, "After, all he taught me to be like him, in whom spiritual life was not separate from any other human interest. What Merton taught me, which I could never have learned in classical mysticism, is that my life was the

only 'spiritual life' I could have, and not any other." (Cardenal 2003, p. 34). This paradox has been pointed out by Jordan (2015), who states that, "Though Cardenal's departure from strict pacifism appears to deny the spiritual and social ideals of his Trappist background, he justified support for the revolution through his social commitment as a contemplative Christian—a commitment he learned from Merton" (8). If careful attention is paid to the epistolar exchange between Merton and Cardenal from 1959 until 1965 (that is, from the moment Cardenal left the trappist monastery up to the time when he founded the community at Solentiname), the connection between poetry, politics, the inner life, mysticism and monasticism is evident. The letters, written in the 1960s, can be read together with Cardenal's memoir *The Strange Islands*, and the image one obtains is that of a deep relation that involves discussion about several "worldly" and literary issues, as well as on mysticism. In the memoir, Cardenal partially quotes the letter he received from Merton, written on the day of his ordination. In the volume edited by Daydí-Tolson, the full letter has been compiled, and it is possible to read Merton's blessings:

> May God bless your priesthood and all your priestly work, especially all the splendid inspirations you have received. May all of them come to fruition. It is true they will not without much difficulty, but it is a happy motive that the Church breathes a new spirit of understanding and originality [ . . . ] Your life has been blessed, your vocation certainly comes from God in the most evident ways. He may let you feel your own limitations, but the power of his Spirit will also be evident in your life. Do not be afraid; be like a child in His arms, and you will do much for your country. (Cardenal and Thomas 2003, pp. 157–58)

In the same letter, Merton makes reference to the six years that have passed since Cardenal left Gethsemani. If attention is paid to the letters exchanged over that period, it becomes evident that he knew about the plans for Solentiname from the very beginning: as early as 1962, Cardenal refers to the possibility of establishing a small community in a quiet place (Cardenal 2003, p. 88), but it is only on 28 January 1965 that Solentiname is mentioned for the first time:

> I have already chosen the place where I wil settle [ . . . ] at the end of the year. It is on an island in the Solentiname archipelago, in the lake of Nicaragua [ . . . ] The bishop has accepted my plan, and he will ordain me. The papal Nuncio has also approved it, and they are enthusiastic about my plans. (Cardenal 2003, p. 148)

In fact, Merton's influence over the foundation at Solentiname was to be so important that before settling down, in the autumn of 1965 Cardenal went on a sort of pilgrimage to Kentucky, to obtain instructions from his former master, in what could be part of the illumination moment in his mysticism. Merton was very clear about how to start at Solentiname: "The first rule is that there be no rules. And after this, then, all other rules are unnecessary." (Cardenal 2003, p. 80). The challenge of existing social order is at the basis of Merton's suggestions for Solentiname, and the way Cardenal organizes his literary discourse in *The Strange Islands* makes it clear that there were no rules, at least not any conventional rules, in his foundation; if there were original ones, they soon tended to be overlooked, as can be seen below.

Cardenal has said that Merton was like a father to him, and when thinking back on his death, he recalls, "Even if Merton was only 10 years my senior, to me he was like a father, and his death was the deepest sorrow I have felt." (Cardenal 2003, p. 235). Solentiname was a Latin American project and praxis, but it would not have existed had it not been by Merton's presence in Cardenal's life. After Merton's death, Cardenal met some of the Trappist's American friends: "at the Merton Center, we talked about how he was to go to Solentiname after the Asian journey. And Dan Berrigan asks me, 'Are you sure he's not there?'" (Cardenal 2003, p. 236).

### 2.1.2. Cardenal's Interest in Central American Indian Culture

From his days at the seminary to his project in Solentiname, Cardenal shows a deep interest in and involvement with Central American Indians. This cannot be separated from Merton's influence: "I have already said that it was a *gringo* who showed me the indians. When I was a novice in a holy United States, Thomas Merton revealed to me the wisdom, spirituality and mysticism of the indians of the Americas." (Cardenal 2003, p. 37). In a letter written very shortly after Cardenal left the novitiate, Merton tells him about the possibility of a monastic foundation, "one should be definitely rooted in the indian and Latin American cultural complex." (Merton and Cardenal 2003, p. 59). Cardenal studied native cultures at the Ethnographic Museum in Bogotá (Cardenal 2003, p. 37), and he admires aspects of that culture which are similar to the monasticism he wishes to live: "They have embraced poverty as a religious order. They say the life of the rich goes against wisdom. [ . . . ] Like trappists, they never say mine or yours." (Cardenal 2003, p. 37). After studying about them, while he was still at the seminary, Cardenal visited the cunas of Colombia, close to Panama, and on another occasion he went to see the ticunas of the Amazon, near Peru. On his way back from Gethsemani, after visiting Merton for instructions, he saw the native people of New Mexico, of whom he admires their simplicity and their prophetic knowledge. Of the former, Cardenal says, "One of them told me men were not happy because they were not satisfied with what they had, they always wanted newer cars [ . . . ] If everyone were satisfied with something, all would have enough." (Cardenal 2003, p. 86). He also quotes a prophecy of the Indians: "On that new earth to come, indians and white people would be brethren, and this is a prophecy yet to be fulfilled." (Cardenal 2003, pp. 87–88). Recalling those words decades after Solentiname, they can be taken as evidence of Cardenal's new understanding of community and of mysticism in relation to the Latin American heritage and context.

### 2.1.3. Latin American Revolutionary thought and Praxis

Even if his family and close friends were strong anti-somocistas, Cardenal seems a detached observer at first. In September 1959, shortly after leaving Gethsemani, he writes to Merton, "My brother Gonzalo is currently head of the clandestine movement in Nicaragua, doing very dangerous activities, like introducing weapons, communicating with foreign revolutionaries or making dynamite bombs [ . . . ] I have just learned this from my brother-in-law [ . . . ] Our prayers are much needed." (Merton and Cardenal 2003, p. 51). Of course right before learning this, he had been at the abbey, where this type of news could not reach him, thus the apparent detachment from the family involvement in "dangerous activities." Yet, once he learns about the revolutionary doings and leaders, his discourse changes, and he shows enthusiasm. Through the news of revolutionaries, Cardenal's interpretation of the Gospel becomes socially involved, and it is possible to see the seeds of what would later be reflected in *The Gospel in Solentiname*. While Cardenal was still at the seminary in Colombia, "Camilo Torres emerged in those days." (Cardenal 2003, p. 63). Ernesto, as well as many others at the seminary and across Colombia, were secretly reading Camilo to "become priests and not declare our camilismo before it was due time." (Cardenal 2003, p. 65). Even if Cardenal had befriended other Marxists or revolutionaries in Mexico or Colombia, "One novelty in Camilo is that he called for the union of marxists and Christians, to fight for the revolution." (Cardenal 2003, p. 65). Looking back, when he writes his memoir Cardenal uses the first-person plural to say, "He [Camilo] did not reach what we would later, that is, the union of Christians and marxists." (Cardenal 2003, p. 65). Camilo Torres' impact on Cardenal's thought and praxis as part of his illumination on the Latin American reality cannot be overlooked: he recounts details of Camilo's texts, participation in the guerrilla movement, and even the fact that he (Camilo) fell only five days after Cardenal and his friends arrived at Solentiname to start the foundation. "He [Camilo] was the first guerrilla priest in Latin America, and an example that many have followed. But he is not just an example to guerrilla priests: with or without guerrilla, his life and death have set an example for everyone, priests and all." (Cardenal 2003, p. 67). Many

years later, when Cardenal was already a public figure, he visited Cuba on an invitation of Casa de las Américas. The visit, described in *In Cuba* (1972), is recalled in *The Strange Islands* as well, and the discourse shows an enthusiasm for revolutionary thought and praxis similar to the one he felt in his youth, when he first heard of Camilo Torres.

2.1.4. The Experience of Mystical Union

As said earlier, while Cardenal was a novice under Merton, he learned that the spiritual life was the life he had, related to all his interests and aspirations. In *The Strange Islands*, he recalls that while at the seminary, he was part of the "Hallelujahs", a group of seminarists who not only shared political views, but also sympathized because of their spirituality, quite opposite to the strong clericalism of their superiors. In the chapter "Un seminario en los Andes" (A seminary in the Andes), he devotes four pages to describing the delights of mystical love in an exalted tone. The imagery he uses is that of traditional mysticism:

> Intimacy with the Inifinite, how can I explain that? It is a union within oneself, and without feeling it with the senses I feel it, his forehead on my forehad, his eyes on my eyes, his mouth on my mouth, so close to me that I no longer know who is who, who I am and who He is, where He begins and I end, because He and I are one, one you only, and one I only, a you that is me and an I that is you. [ . . . ] In my room in front of the Andes I could feel He invaded me and embraced all my being, body and soul, satisfying all the desires of my soul and of my body [ . . . ]. (Cardenal 2003, p. 30)

The joys of mystical union are such that language seems insufficient to convey them; the logic of ordinary language is surpassed by the beauty of absolute satisfaction brought on the soul by God, who fills the vacuum left by the absence of human love. Cardenal's mystic literature has been studied, among others, by García González (2011). García González describes the poet's efforts to convey the consequences of his mystic revelation or experience of 2 June 1956, narrated in *Vida perdida* (*Lost Life*) (Cardenal 1999), the first volume of his memoirs). García González concentrates on Cardenal's poetry (which we do not tackle in this article) and in what she calls his works "of mystic theme" (p. 49), that is, texts where the sociopolitical concerns are not apparent. Yet, as can be seen in his memoirs, both the rhythm and enthusiasm, the love imagery and the forcing of language to convey in as much as possible the ineffable, go beyond poetry, and the traits of mystic discourse permeate his prose as well.

In Cardenal's discourse, one may detect that love imagery, which is at first physical and sexual, and turns into a contemplative mood when he describes the overwhelming beauty of the natural world at Solentiname:

> These are all figures of love, I said. Observing what surrounds us was being in a dialogue with God. [ . . . ] And I say also that God speaks to us when we are hearing at night that continuous jua! jua! jua! from the lake, which reminds us who made the lake and these stony islands where we are, and the planet on which the lake is, and the whole universe. Which is, by the way, the very same one inside us. (Cardenal 2003, p. 109)

In this way, Cardenal is able to show that the mystical union is not only between the soul and God, but between human beings and creation.

Based, then, on the advice he sought from Merton and on what he had learned from the Trappist monk, rooting himself in his native Nicaragua, slowly stepping from an almost purely contemplative to an engaged stand, Cardenal starts his community at Solentiname as a site of liberation from established rules, orthodox biblical hermeneutics, worldly ties and, after all, from social injustice and political oppression as well. Through his contacts with revolutionary leaders, Cardenal changed not only from his detached observation of reality in 1959, but also from his pacifist views to an understanding of the need for engagement with the revolution. In this sense, he represents the type of liberation that Dussel defines as critical of the established order. Cardenal shows how he was gradually

brought into the Sandinista Front and recalls an interview with Eduardo Contreras, Zero Commandant: "The first time I saw him with my brother Fernando [who was a Jesuit priest], and he said, 'We are revolutionaries.' He explained to me that they did not aspire to just overthrow a tyrant, but to change the capitalist system. And then I was convinced." (Cardenal 2003, p. 224). The revolutionary itinerary has been described by Drozdowicz, who shows that Solentiname went from being a contemplative community to engaging with the oppressed society of which it formed a part: "Contemplation and working with the young made [Cardenal] more radical; consequently, a revolutionary conscience grows in Solentiname, and this is why several of its members would later take arms and revolt against Somocismo." (Drozdowicz 2018, p. 168).

In this context, even the flexible rules by which the community lived at Solentiname could be changed, as part of the liberation process. Cardenal as a Latin American mystic cannot be isolated from his community and the demands of solidarity from individuals and society. As Scannone suggests, the injustice suffered by victims puts traditional religiosity in interdiction and calls for new ways. Cardenal makes plenty of references to this; one case is that of Elbis, a young member of the community:

> Soon we incorporated Elbis (he wrote his name with a b thinking it was spelt that way). [ ... ] I had thought candidates for the contemplative life would arrive at Solentiname from other parts, but they did not. I received the Solentiname peasants. [ ... ] Elbis was humble, quiet, loving. Especially with young children. [ ... ] His mother Natalia said he had been a martyr for the children, he told her how the children's suffering made him suffer; he wanted a Nicaragua where children could be happy; and that was mainly what brought him to the revolution in which he died. (Cardenal 2003, pp. 206, 210)

In the Latin American revolutionary context, martyrdom is not just an act of faith and witness, but life offered for the oppressed, as Cardenal shows by presenting Elbis' case. The connection between the mystic way and historicity as Boff and others understand it throws light into this reading of the martyrs and saints that Cardenal includes in his narrative. As a priest, and because of his religious formation, Cardenal and his community read the lives of the saints, but he makes it clear that Solentiname has its own heroes and martyrs (Cardenal 2003, p. 233).

The progressive radicalization of Cardenal's community is linked to its being rooted, as Merton had suggested, in Latin American reality. Once he had started his contact with the FSLN, Cardenal remembers his master's teachings: "I always remembered what Merton told me when we talked about the contemplative foundation we wished to make: that the contemplative should not be indifferent to the social and political problems of his people. Especially in Latin America, where there was so much social injustice and frequently dictatorships as well." (Cardenal 2003, pp. 203–04). Merton's teachings are at the root of the foundation, and Cardenal also explains that he had always felt inclined to worry about social and political issues. "The contact with the poverty of Solentiname peasants, and the ever-worsening national reality also contributed to the politicization and radicalization in myself and in the community. We were getting more left-wing." (Cardenal 2003, p. 204).

### 2.2. A Church and a Gospel of Liberation

### 2.2.1. A Peasant Community Reinterprets the Gospel

The human wisdom that Scannone sees in Latin American people is the essence of Cardenal's listening and learning. In the modern world, biblical hermeneutics has been traditionally owned by clerics. Yet, the Gospel as read and understood in Solentiname became not only a source of popular interpretation but also an opportunity for Cardenal's mystical silence. Silence allows for the soul to hear God's message. In Solentiname, not only God but also peasants speak, while Cardenal listens. The praxis of liberation in the community at Solentiname involves understanding the biblical text based on the surrounding context, to reach new conclusions through different methods, as Dussel suggests.

Sergio Ramírez (2015) explains:

The idea of organizing a contemplative community in Soletiname did not prosper; what did, instead, was a peasant community, based on its commitment to a liberating gospel, not far from marxism. After the Eucharistic Congress in Medellin in 1968, and the breech opened by the Second Vatican Council, Latin American priests and lay people supported the idea of a church engaged with the poor, which in turn led to liberation theology. (Ramírez: caratula.net accessed on 10 March 2023)

The connection between mysticism and politics that scholars have marked as typical of the Latin American experience can be seen fully at play in Solentiname. Yet, Cardenal's method was not his own invention. He gives credit for it to Father De la Jara, a Spanish priest who was "parishioner in a poor barrio where they had this movement called God's Family, a sort of community of couples." (Cardenal 2003, p. 195). De la Jara visited Solentiname and helped create a similar movement. Cardenal implicitly refers to him as a master who "taught" him "not to give a sermon about the Gospel, but a dialogue about it instead, commenting it among all present." (Cardenal 2003, p. 196). In turn, Father De la Jara had learned this in Panama, in San Miguelito, a poor parish "famous for the comments of the Gospel they made there, and they had learned that from a poor parish in Chicago." (Cardenal 2003, p. 196). Making communal comments on the Gospel, then, is not something typical of Solentiname, but a practice that was already at work in the Catholic church in the Americas. Cardenal shows this continental line when he says, "From Chicago this passed on to San Miguelito, and from there to Father De la Jara's parish, which was also famous, and from there to Solentiname, where it produced the book of comments on the Gospel which I would publish later, under the name *The Gospel in Solentiname.*" (Cardenal 2003, p. 196). The community of families and the shared reading of the Gospel existed within an unorthodox Mass liturgy.

Cardenal learned from De la Jara and other priests across the Americas to listen to his community as part of this liberation praxis: liberation from orthodox liturgy, from orthodox Church teaching, and freedom for people to express their ideas and interpretation of the Gospel, which would eventually lead to political liberation as well. Latin American mysticism at its best is communal theology rooted in people's reality. This can also be related to Thomas Merton's influence, who, as quoted above, taught Cardenal that contemplatives should not be indifferent to the social and political issues of their people, especially in Latin America (Cardenal 2003, p. 203).

In the "Introduction" to *The Gospel in Soletiname* Cardenal equates the comments he has recorded in this book with the way the Gospel was composed: "The *peasant's* comments are deeper than those of many theologians, but as simple as the Gospel. No wonder: the gospel or 'good news' (the good news for the poor) was written for them, and by people like them." (Cardenal 1979a, p. 9). The simplicity of style and the wisdom the comments manifest are part of the Spirit that speaks in them: "the true author is the Spirit who has inspired these comments (the Solentiname peasants know well that He makes them speak) and it is the same who inspired the gospels" (Cardenal 1979a, p. 9). The basic human wisdom or popular wisdom that Scannone (2009) (p. 69) defines is seen at play in the comments of Cardenal's community members. As editor of the comments, Cardenal does not overlook who says what, but instead pays close attention to each individual's identity, personality and background, and how all this shapes the comments the Spirit inspires unto them: "The Holy Spirit, which is God's spirit infused in the community, which Oscar would call the spirit of union in the community, and Alejandro the spirit of brotherly service, and Elbis the spirit of future society, and Felipe the spirit of the proletarians' struggle, and Julio the spirit of equality and common property, and Laureano the spirit of revolution, and Rebeca the spirit of love." (Cardenal 1979a, p. 10). The deep union between the Spirit and the people shows both in style and in Cardenal's presentation that his community has achieved communication with the divine and, in this way, mystic communion that leads to liberation.

2.2.2. The Journey towards Liberation

The idea of a journey towards liberation has shaped Judeo-Christian history and identity. Starting with Moses, who led the people of Israel across the desert towards the Promised Land, journeying has been intrinsically joined to the liberation process. God has a historical presence (García 1987, p. 12) among His people, and the mystic realizes that God still walks next to them in our own contemporary times (Mella 2009, p. 389). Whether there is geographical displacement or if the journey is just a metaphor, in both cases liberation implies walking towards it, advancing in life to reach that end.

It is interesting that Cardenal notices in the "Introduction" to *The Gospel in Solentiname* that the comments have not been edited in the chronological order in which they were made and recorded, but according to the chronology of Jesus' life instead. Thus, the revolutionary itinerary cannot be clearly detected in its evolution, but choice of the order of the Gospels (Cardenal 1979a, p. 9) instead shows how liberation was always present as a main concern in Cardenal's foundation.

As said above, Mella (2009) identifies six steps in the liberation process, departing from deep interior silence, as seen in the poetry of Buelta. In the case of Cardenal's text, considering that *The Gospel in Solentiname* is a polyphonic text (because the voices of all participants in the comments of the Gospel are included, transcribed from tape recordings), the steps of the individual mystic path should be extrapolated to a communal experience.

1.      Loss of meaning of known discourse.

Dussel points out that in order for the new system to be born, the old one must pass away (Dussel 2013, p. 83). Read in terms of discourse, this implies that the old message, the old words, no longer possess meaning, and realizing this is the first step towards change. In Cardenal's community, the change takes place both at the level of religious practice and of social-political involvement. Commenting on the episode of Nicodemus' visit (Jn 3, 1–21), Olivia says, "And in spite of what Jesus said, we still have that religion of not eating meat on Friday, and no one cares if a poor is killed that day! That the candle is lit to pray the rosary, but if people are hungry, that is God's will! So that is why Christ told them that could not happen. It is better to fight against injustice than to be with that false religion [ . . . ] as many people still do. A lot of people who fast and have hard hearts." (Cardenal 1979b, p. 20).

It can be argued that throughout *The Gospel in Solentiname*, "known discourse", or the traditional interpretation of the Gospels, has lost meaning, thus also lost are the necessity of the comments and of this book by Cardenal. That loss of meaning calls for a new understanding of the biblical message.

2.      Invitation to search new social meanings.

While traditional theology and hermeneutics may see a divide between earthly concerns and the religious, liberation theologians overcome the gaps between the various realms of human experience, and they "can meaningfully raise the question of the relationship that exists between the creation of a more just world and the kingdom of peace and justice" (García 1987, p. 17).

By having the type of Masses Cardenal describes in *The Strange Islands* as quoted above, in which dialogue leads to constructing new interpretations, he invites the people in his community to search social meanings in the biblical text. When discussing the Prologue to John's gospel, the priest and the peasants comment:

Me: God became man, so now man is God. [ . . . ] The word now is the people. The people now do God's work.

Felipe: Without the need for God to do it.

Me: God has nothing to do here now. He began the work of creation, but now he has left that to ma, so that man keeps doing it.

Oscar: As God has nothing to do here, it seems we have a huge responsibility in mending the world, so that those who are away from him and are not His



children will be convinced by us and will be His children too, and we'll be united as brethren. That is our fight: to be all one.

Laureano: We have to mend the world, establish justice on earth, make revolution. (Cardenal 1979a, p. 15)

Even if God "has nothing to do" here and now because He has passed on the responsibility to His people, the message is not totally secular or totally social. The new social meanings that are being created keep links with God because God himself became one of us, thus the need for a social meaning attached to the Gospel.

3. Understanding that God still walks next to the people.

All throughout *The Gospel in Solentiname* there are accounts of the ways in which people of the community feel their own experience is like that narrated in the Gospels. However, their words show not only a sort of imitation or of being like mirrors or duplications of stories that happened in the past, but instead there is an understanding that they are part of the people of God. Towards the end of the second volume, when they comment on the Resurrection, Cardenal tells his parishioners:

Me: It is true that they [the official church] have put the resurrected Jesus in heaven, in another life, in the afterlife, so that the earth does not change and there is still injustice and poverty. But he resurrected to be on earth: 'He was dead and is on his way to Galilee before you.' And he is here in Solentiname too. And it's curious how for the first time he calls the disciples, 'brothers.' Before he had said, 'My Father and your Father.' Now, after resurrection, he calls them brothers.

Esperanza: He is a *guerrillero*. He was killed for liberation. All who fight for liberation and die for it and resurrect are his brothers. (Cardenal 1979b, p. 295)

In this way, the struggles of the Nicaraguan Sandinista movement and the revolutionary plight in Solentiname are not only a reflection of Jesus' and the disciples' lives, but a true continuation in the same line, with Jesus walking side by side with them.

4. Looking back on history to understand that God has always manifested Himself in liberation processes, while never ceasing to be transcendent.

Just as this mystic people feel God is with them, there is a realization that He has been present at all moments, whenever and wherever there has been a liberation movement:

William says, In the Bible, the Almighty God had always revealed Himself as the liberator of his people. He manifested Himself first with Moses, who f . . . the Pharaoh. And then, through the prophets, He fought all types of oppression. His son, this Jesus, the Yahve-liberates, will be like him. And he will be king.

Oscar: The angel announces a new government with him. It is the kingdom of the poor. This kingdom is being set from the time Christ came to earth, but it is not fully established yet.

Don Julio: I'd say it's just beginning. (Cardenal 1979a, p. 18)

God has manifested himself against all types of oppression, spiritual and social, political and religious, without ceasing to be God, speaking through the prophets and sending His own Son as part of the liberation scheme which has not finished yet.

5. A new way of understanding one's own commitment to freedom as the fragile manifestation of what really matters.

Cardenal and the community at Solentiname grow aware, by degrees, as is proved later, that they are part of a new church, a new society in which they will manifest Christ, just as the ancient prophets:

Joining Christ is joining everyone. It is joining the community. Through a small community, we are united to a bigger community, like the small branches are united to bigger ones. And all branches together is Christ. (Cardenal 1979b, p. 248)

6. At the end of the journey (the mystic journey), a deep peace enables the mystic to accept he/she is a regular human being, whose achievements in the struggle for liberation are not a result of his/her own efforts, but " a humble participation in divine life, which is given graciously everywhere, without asking for anything in return". (Mella 2009, p. 382).

According to the way Mella poses this idea, the liberation process is not only part of a social struggle, but intrinsically connected with the spiritual liberation of all human beings. Cardenal is aware of the radicalization of his own message and of the interpretations of the Gospel the community make as time passes by. Yet, this does not mean they become secular; on the contrary, they never miss the connection with God:

I say, Politics in the gospel is the communion of all men, who share all things, and to do this, a new birth is needed. We must leave behind the old man (the man of the old society), St. Paul says, and dress ourselves in the new man, without distinctions among Jewish or Greek, lords or slaves. [ . . . ] Che himself dressed like this new man a lot.

[ . . . ]

Oscar: I don't see why keep talking about heaven, wishing to go up to heaven now, I believe there's enough to see here on earth.

Olivia: I think the things of earth are the same as the things of heaven.

A girl: When people love each other there is a community of love, and that is heaven: where there is no division, no selfishness, no falseness, there is heaven, that is heaven, that is glory . . . (Cardenal 1979b, p. 22)

This comment about Nicodemus' visit (Jn 3, 13–21) summarizes the mystic journey and the inner and communal peace that can be achieved even in the midst of the struggle for liberation. The liberation journey, and the mystic journey, are communal experiences in which the delights of heaven can be sensed on earth as well.

### 2.2.3. A Social and Political Reading of Liberation

As Andiñach and Botta (2009) point out, new ways of reading and interpreting the Bible stemmed from the social injustices of Latin America, "as a demand from a reality that, when confronting the biblical text with an experience of oppression and subjugation of human rights, imposed a reading that would privilege sense and message" (p. 5). In Latin American liberation theology of the 1970s, "there was a firm purpose in both the general theological field and biblical theology in favor of getting involved with the social sciences in order first to understand reality and then to commit to the struggles to modify it so that oppression and injustices may be overcome" (Andiñach and Botta 2009, pp. 5, 6).

The progressive deepening of revolutionary concerns is underlined in *The Strange Islands*. In the chapter "Y hasta las sardinas parecen cantar" ("And even the sardines seem to sing")[3], Cardenal recalls the comments about the episode when Jesus calmed the tempest (Mk 4, 35–41), quoting quite literally from *The Gospel in Solentiname*. One of the peasants, Bosco, comments that repression is the danger they face, and Cosme answers that, "We are undergoing a rain of injustices. Inequalities are the waves going up and down", to which Olivia adds, "He travels with us in the community. The boat is the community." (Cardenal 2003, p. 246). Cardenal explains, "As can be seen in these comments, we were already in an unquiet climate. Those were the times when the revolution was approaching, as I said." (Cardenal 2003, p. 246). In *The Gospel in Solentiname*, the same conversation, recorded and transcribed, includes passages such as the following: "Felipe: Faith is what many youngester have nowadays, faith in change, in the revolution. It is faith in that the world may change with love, evil can become good, those brave waves can be calmed down." (Cardenal 1979a, p. 229).

On the other hand, in the episode of the Wise Men visiting the baby Jesus (Mth 2, 1–12), for example, Cardenal himself opens the dialogue saying, "As way of introduction, I say

that when Matthew says, 'in the time of King Herod', he is saying that Jesus was born under tyranny. There were three Herods, like saying three Somozas in Nicaragua." (Cardenal 1979a, p. 40). Even the fear that Nicaraguans feel when they think of overthrowing the dictator is equated to the attitude of the Wise Men who consult Herod before visiting the infant Jesus (Cardenal 1979a, p. 41). In the same chapter, Olivia reflects that when Mary, the mother, "was pregnant, she had sung that her son would dethrone the mighty and give riches to the poor, and leave nothing to the rich." (Cardenal 1979a, p. 43). The next episode from the Gospel included in the book is the killing of the innocent (Mth 2, 12–23), and the political violence in Nicaragua has evidently suffered an escalade, because the comment is more overtly political, revolutionary and denouncing than the previous chapter. Cardenal explains, "Little before Mass this Sunday, a patrol came inspecting our houses (the country is under martial law and individual guarantees have been suppressed). Some people seem to be scared[ . . . ]" (Cardenal 1979a, p. 44). Such political circumstances lead the commentators to speak of Jesus as a subversive, as Cardenal's brother Fernando (himself a Jesuit priest) who was visiting Solentiname at the time, says: "[Mary] realized she had given birth to a subversive messiah [ . . . ] And I think for a long time we have been misreading the gospel, interpreting it from a purely spiritual perspective, overlooking all its political and social circumstances." (Cardenal 1979a, p. 44).

Throughout *The Gospel in Solentiname*, the peasants reflect based on their own sociopolitical circumstance, and those who have had theological, philosophical or political training reconsider and reinterpret the teachings they received before being involved in the theology of liberation. In this sense, the novelty of the movement can be found in what Ismael García defines as "its political option for the poor, making them and their struggle a focus from which to engage in meaningful theological reflection" (García 1987, p. 8).

In the new approach that Latin American mysticism of liberation brings about, the message is still connected to the spiritual, but in its deep connection with the social and the political. In any case, neither of the aspects should be overlooked, because as Argüello states, "Solentiname was not simply a cultural or political project; the deep meaning of that experience is theological: through Cardenal's prophetic priesthood, God himself was present among the peasants in a remote corner of Nicaragua, and He penetrated the history of our martyred and oppressed people" (Argüello 1985, pp. 365–66). This presence of God among people is part of the dialectics of the theology of liberation, in which García explains "a hermeneutical circle between the sociopolitical and historical praxis of the community of faith and its interpretation of Scripture, the theological tradition, and in particular its interpretation of God's historical presence" (García 1987, p. 12). Ernesto Cardenal, his brother Fernando and other visiting priests and intellectuals make comments rooted in traditional theology; yet, once they are committed to the struggle for sociopolitical liberation in a context where God is seen in full presence among His people, the mere interpretation of Scripture changes, and the Bible is re-read anchored in the circumstances of the reading/interpreting moment, which challenge the traditional thought in which Cardenal and his fellow priests and scholars had been trained.

## 3. Conclusions

Following Londoño (2021), we may say that Cardenal's discourse shows how his mysticism (and that imprinted on his Solentiname community) shows a political stand, where liberation is not just, or not mainly, a spiritual notion but a way to become politically engaged. Social justice is at the center of Cardenal's community in Solentiname. Even if in the plans prior to settling on the archipelago he might have had in mind a purely contemplative foundation, reality soon led him to put into thought and practice the basis of liberation theology, for which the spiritual cannot be separated from the sociopolitical, economic, cultural and other realms of human experience.

The revolutionary commitment that Enrique Dussel sees in Cardenal is evident in the way he and especially the community at Solentiname comment on the gospels, feeling a very deep identification with every story and character from the Bible. Jesus' humble origin,

his upbringing in a family of artisans, his role as a leader, the sufferings of his crucifixion and his final resurrection/liberation are interpreted by the commentators in Cardenal's foundation in the light of their own liberation struggle. The mystical encounter with God is perceived not only in Cardenal himself, as an individual, but in his community. The Solentiname peasants and their visitors alike realize that God really walks next to them in the struggle and journey towards liberation. Read half a century after they were recorded, the comments of the Gospels in Cardenal's *The Gospel in Solentiname* resonate in their commitment and identification with a concrete historical and sociopolitical circumstance, that of the preparatory stages of the Nicaraguan revolution, in a clear manifestation of God's presence in history as understood by the theology of liberation. In this sense, then, Cardenal and his community experience a communion among themselves and with God, so that the community is truly mystical.

Ernesto Cardenal is a mystic in the traditional sense, marginalized (geographically in Solentiname, away from the centers of power, and institutionally, because the official Church did not look favorably upon him or his foundation), though he holds spiritual authority in his community, and lives a deep communion with the divine in a spiritual and communal sense. At the same time, Cardenal is a mystic in the Latin American tradition of liberation, in which the social and political circumstances cannot be overlooked, because they lead him to political engagement. The deep communion with the divine which has characterized various mystic traditions across the world is, in Cardenal's Latin American liberationist trend, inseparable from community with the poor and the oppressed who fight for justice. As a mystic and a prophet, Cardenal also had a political and revolutionary mission: Solentiname became Cardenal's own a praxis of spirituality and liberation in the context of Latin American social movements of the last quarter of the 20th century. Deprived of its mystic sense, it would be incomplete: Cardenal (and his community) struggled for liberation from economic and political oppression, but the spiritual aspect of society and the need for union with God was never overlooked or secondary to their concerns.

**Funding:** This research was funded by CONICET during the biannual research perios 2020–2022.

**Institutional Review Board Statement:** Not applicable.

**Informed Consent Statement:** Not applicable.

**Data Availability Statement:** Not applicable.

**Conflicts of Interest:** The author declares no conflict of interest.

## Notes

[1] I used the original texts in Spanish for this article, and prepared my own translations. Yet, for the sake of fluent reading, whenever the titles are included in the text, they are mentioned in English, as *The Gospel in Solentiname* and *The Strange Islands*.

[2] Authors' translation of all the citations from bibliography in Spanish.

[3] The title is the line from one of the songs in *La misa campesina nicaragüense*, by Carlos Mejía Godoy and Oscar Gómez (1979).

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
