# Peer review of "Ernesto Cardenal: A Latin American Liberation Mystic"

_religions, doi:10.3390/rel14050655_

Round 1

Reviewer 1 Report

This an interesting but at the same time variously problematic essay.

The text is too long. There are many repetitions in it. The structure is not fully clear because the mysticism of Cardenal is not obvious from many of the quotes in the text. There are a high number of typos. The use of book titles is confused in term of language. The translations of the Spanish texts are not always well-done.

What to do:

The author has to revise the entire text grammatically and formally (eliminating typos and grammatical problems). Spanish words are to be used in translated texts if and only if they cannot be properly translated. The text must be shortened with a focus on the mystical elements in Cardenal’s oeuvre. The titles of Cardenal’s works are used sometimes in English, sometimes in Spanish, which is not right. A stronger focus on the argument is needed. In the conclusion, the aim of the paper is well stated, but in the main text, this thesis is blurred. 

Too many typos, untranslated Spanish words. 

Reviewer 2 Report

It was appreciated that the author starts immediately by clarifying the Latin American philosophical underpinnings and context for Cardenal's experience and work.

Clear and logical structure. Easy to follow.

It would be preferable to change the title of the paper so it reflects the more focused content of the contribution.

Sometimes the author relies heavily on Enrique Dussell and in some instances the article is descriptive, mainly elucidating citations from Cardenal's text. A more critical approach would be appreciated.

Overall , quality of English is good.

However text should be revised for misprints. The following were noted:

218 all the

278 will

293 only

426 of

483 famous

497 smoked, scandalised

532 Judeo-Christian

539 The Gospel of Solentiname

582 man

620 f (?)

730 traditional

there might be other misprints. Kindly check and revise the text also for consistency in italicisations.

Reviewer 3 Report

Overall good assessment of the place of Ernesto Cardenal within Latin American liberation theology and contextual mysticism. The article traces his growth as a thinker, theologian and revolutionary through the influences of Fr. De la Jara, Thomas Merton as affirmed through the sources of Boff, Dussel and especially his own autobiographical writings. One question that I have is how Cardenal was able to reconcile mysticism within his own spirituality with the violent action within Camilo Torres' liberation theology? 

The name of "Moses" is mispelled about half-way through the article (line 533). Also, should the "Gospel According to Solentiname" be italicized as the title of a book (see line 539)? 

Round 2

Reviewer 1 Report

The text is better now but there are still weird mistakes in spelling, for instance, a reference to "DDD" (line 417) which is obviously an error. In the Bibliography, there is a line beginning with 1 (line 778). Otherwise, the amends are fine. 

The English is improved a little, it is in order now in my judgment. 

Author Response

Dear reviewer,

I have corrected the name that was missing in line 417; I have erased the number 1 that got into the bibliography, and I have made very minor changes in structure where it made the sentences more fluent.